# Tolerogenic IDO1^+^CD83^−^ Langerhans Cells in Sentinel Lymph Nodes of Patients with Melanoma

**DOI:** 10.3390/ijms23073441

**Published:** 2022-03-22

**Authors:** Gianni Gerlini, Paola Di Gennaro, Nicola Pimpinelli, Serena Sestini, Lorenzo Borgognoni

**Affiliations:** 1Plastic and Reconstructive Surgery Unit, Regional Melanoma Referral Center and Melanoma & Skin Cancer Unit, Santa Maria Annunziata Hospital, 50012 Florence, Italy; gianni.gerlini@uslcentro.toscana.it (G.G.); paola.digennaro@unifi.it (P.D.G.); serena.sestini@uslcentro.toscana.it (S.S.); 2Department of Health Sciences, Section of Dermatology, University of Florence, 50125 Florence, Italy; nicola.pimpinelli@unifi.it

**Keywords:** melanoma, Langerhans cells, indoleamine 2, 3-dioxygenase, sentinel lymph node, tolerance

## Abstract

Langerhans cells (LCs) are crucial regulators of anti-cancer immune responses. Cancer, however, can alter DCs functions leading to tolerance. The enzyme indoleamine 2,3-dioxygenase (IDO1) plays a crucial role in this process. In sentinel lymph nodes (SLNs) of patients with melanoma, LCs show phenotypical and functional alterations favoring tolerance. Herein we aimed to investigate IDO1 expression in SLN LCs from patients with melanoma. We showed by immunofluorescence analysis that a portion of Langerin^+^ LCs, located in the SLN T cell-rich area, displayed the typical dendritic morphology and expressed IDO1. There was no significant difference in the expression of IDO between SLN with or without metastases. Double IDO1/CD83 staining identified four LCs subsets: real mature IDO1^−^CD83^+^ LCs; real immature IDO1^−^CD83^−^ LCs; tolerogenic mature IDO1^+^CD83^+^ LCs; tolerogenic immature IDO1^+^CD83^−^ LCs. The latter subset was significantly increased in metastatic SLNs as compared to negative ones (*p* < 0.05), and in SLN LCs of patients with mitotic rate (MR) > 1 in primary melanoma, as compared to MR ≤ 1 (*p* < 0.05). Finally, immature SLN LCs, after in vitro stimulation by inflammatory cytokines, acquired a maturation profile by CD83 up-regulation. These results provide new input for immunotherapeutic approaches targeting in vivo LC of patients with melanoma.

## 1. Introduction

According to the cancer immune-editing theory, tumor cells are able to evade the immune system by escape mechanisms responsible of tumor tolerance [1]. Among skin tumors, melanoma is considered the most tolerogenic one by its ability to alter immune cells functions, particularly within the sentinel lymph node (SLN), the first draining node from a tumor region, favoring SLN early metastases [2,3]. Melanoma cells create an immunosuppressive microenvironment by cytokines secretion, such as IL-10 and TGF-beta [4,5]. This melanoma-induced milieu is responsible for several morphological, immunophenotypical and functional alterations in SLN dendritic cells (DCs) of patient with melanoma, [6,7,8], as compared to DCs from healthy LN [9]. A particular DCs subset, plasmacytoid DCs (pDCs), displays functional defects in SLN of melanoma patients and accumulates in LNs with melanoma metastases [10]. Even SLN Langerhans cells (LCs), the typical DC subset of epithelia, show an immature immunophenotype, with low levels of CD83 expression, the DC maturation marker, particularly in melanoma bearing SLNs [11]. Additionally, SLN LCs are functionally defective in inducing T cell proliferation and cytokine release [12]. In the past years, the studies on tumor immune escape mechanisms led to the identification of indoleamine 2,3-dioxygenase (IDO) enzymes (IDO1 and 2), responsible of tryptophan (TRP) degradation and kynurenines (KYN) production, as key factors in tumor induced immunosuppression by direct and indirect effects on T cells [13,14]. Particularly, TRP depletion and KYN generation, through aryl hydrocarbon receptor (AhR), induce suppression of effector T cells and generation of T regulatory cells (Treg) by FOXP3 induction in the tumor microenvironment [13,14]. Consistently, Treg counts are significantly increased in peripheral blood of late-stage melanoma patients and positive SLNs [15,16,17], mediating immunosuppression mainly through adenosine generation [17].

In patients with melanoma, IDO1 expression were first observed in a subset of cells with plasmacytoid shape in SLN [18,19]. Some years later, these IDO1^+^ cells had been identified as pDC [20] by the specific marker BDCA-2 [21] in SLN of patients with melanoma. The presence of numerous IDO1^+^ pDCs in metastatic SLN suggested a correlation with substantially worse clinical prognosis in these patients [20]. Another skin derived DC subset, SLN FXIII^+^ dermal DCs (dDCs) express IDO1, probably induced by melanoma-derived TGFb-2, and able to transform peripheral DC into tolerogenic cells [5,22]. Consistently, a negative prognostic role of IDO1 expression in SLN has been reported [23], paving the way for the therapeutic use of IDO1 inhibitors to revert tumors tolerance [13,24,25]. In addition, primary and metastatic melanoma cells can express IDO1 [26,27]. IDO1 expression in SLN LC has not been investigated yet. In healthy human skin, immature LCs do not express IDO1, while after in vitro-induced maturation, IDO1 is expressed by a fraction of mature CD83^+^ LCs, suggesting a potential regulatory/inhibitory function of this subset [28,29]. Since LCs are a promising target for cancer immunotherapy [22,30] we aimed to investigate the expression of IDO1 in SLN LCs of patients with melanoma, correlating it with CD83 and with clinical features of patients. Finally, we investigated the possibility to modulate SLN LC CD83 and IDO1 expression.

## 2. Results

### 2.1. Clinical Features of Patients with Melanoma

A total of *n* = 50 SLNs from *n* = 44 patients were investigated as shown in Table 1. Metastases were detected in *n* = 10 SLNs, indicated as positive SLN, from 8 patients (stage III; 4 women and 4 men; mean age: 62 years, range 27–80 years; average Breslow thickness: 4.19 mm, range 1.1–9.8 mm; mean interval between primary melanoma excision and SLN biopsy: 39 days, range 0–61 days). SLN without metastases were *n* = 40, indicated as negative SLN, from 36 patients (stage I/II; 20 women and 16 men; mean age: 55 years, range 21–93 years; average Breslow thickness: 2.18 mm, range 0.5–8.1 mm; mean interval between primary melanoma excision and SLN biopsy: 41 days, range 0–71 days).

### 2.2. Localization of LCs within SLN of Patients with Melanoma

Since DCs in SLNs of patients with melanoma have been shown to have significant morphological defects, we first investigated LCs morphology and localization within the different SLN areas in frozen sections. By the use of specific Abs for LCs, B and T cells, we observed that Langerin^+^ LCs accumulate in CD3^+^ T cells-areas of SLNs (Figure 1), without differences between negative and positive SLNs. SLN LCs displayed the typical LCs morphology with thin and long dendrites, (Figure 1, inset).

### 2.3. IDO1 Expression in SLN LC of Patients with Melanoma

We first investigated by fluorescent immunocytochemistry IDO1 expression in SLN LCs, the first DC subset to encounter cutaneous primary melanomas. Langerin^+^ LCs were detected in both negative (*n* = 10) and positive (*n* = 4) SLNs from *n* = 12 patients. A few Langerin^+^ LCs co-expressed IDO1 (Figure 2a,b). We detected also Langerin− IDO1^+^ cells, characterized by different sizes, with a predominance in positive SLNs as compared to negative ones (Figure 2a,b). Flow cytometry analyses showed that Langerin^+^ LCs represented a low percentage of total cells, with no differences between negative (*n* = 21) and positive (*n* = 5) SLNs (0.30%, range 0.10–0.72%, and 0.27%, range 0.21–0.46%, respectively; *p* = 0.68) (Figure 2c,e). A portion of Langerin^+^ LCs co-expressed IDO1 in both negative and positive SLNs (3.63%, range 1.32–5.56%, and 4.18%, range 2.60–7.22%, respectively; *p* = 0.53), with a higher percentage in positive SLNs although differences were not significant (Figure 2d,f). The frequency of mature CD83^+^ LCs was lower in positive SLNs as compared to negative ones (30.33%, range 8.75–46.04%, and 40.51%, range 12.98–68.08%, respectively; *p* = 0.27, Figure 2g,h) in line with previous report [11].

### 2.4. IDO1 and CD83 Concomitant Expression Analysis Identified Four SLN LCs Subsets

Next, we investigated IDO1 and CD83 co-expression in SLN LCs from patients with melanoma. By flow cytometry analyses, four subsets of Langerin^+^ LCs were detected in both negative (*n* = 21) and positive (*n* = 5) SLNs from *n* = 22 patients: IDO1-CD83^+^ LCs (named as real mature LCs); IDO1-CD83- LCs (named as real immature LCs); IDO1^+^CD83^+^ LCs (named as tolerogenic mature LCs, most likely representing the regulatory/inhibitory subset observed in the skin [29,31]; and IDO1^+^CD83^−^ LCs (named as tolerogenic immature LCs) (Figure 3a–c). Correlation of the different subsets with the status of SLN revealed that the tolerogenic immature LCs were significantly higher in positive SLN as compared to negative ones (22.01%, range 8.35–46.39%, and 7.00%, range 0.75–21.57%, respectively; *p* < 0.05) (Figure 3d). Light differences, although no significant, were found for real mature LCs, lower in positive SLN (34.42%, range 10.27–51.63%, and 45.32%, range 9.57–71.67%, respectively; *p* = 0.27), for the real immature LCs, lower in positive SLN (37.51%, range 28.57–43.23%, and 44.56%, range 17.93–80.01%, respectively; *p* = 0.39) and for the tolerogenic mature LCs, higher in positive SLN (3.48%, range 1.71–7.98%, and 3.32%, range 0.87–7.37%, respectively; *p* = 0.89). (Figure 3d). Next, we correlated the different SLN LCs subsets with Breslow’s thickness and mitotic rate (MR) of primary melanoma. Concerning Breslow’s thickness, within the 22 melanoma patients enrolled for flow cytometry analyses, *n* = 6 had thin melanomas (<1 mm; all negative SLNs), *n* = 9 had intermediate melanomas (>1 mm and <4 mm; *n* = 7 negative and *n* = 2 positive SLNs), and *n* = 7 had thick melanomas (>4 mm; *n* = 5 negative and *n* = 2 positive SLNs), but no differences were observed in SLN LC subsets frequencies (Figure 3e). Taking into account the MR of primary lesions, *n* = 9 had mitosis ≤1 (all negative SLNs), *n* = 6 had mitosis >1 and <5 (*n* = 4 negative and *n* = 2 positive SLNs), and *n* = 7 had mitosis ≥5 (*n* = 5 negative and *n* = 2 positive SLNs). Notably, the frequency of the tolerogenic immature SLN LCs subset was significantly increased in patients with intermediate (>1 and <5) and high (≥5) MR of the primary lesion, as compared to patients with low MR (≤1) (10.52%, range 3.55–20.95%; 19.31%, range 7.42–46.39%; and 3.25%, range 1.21–6.39%, respectively; *p* < 0.05), while no significant differences were observed between intermediate and high MR (*p* = 0.30) (Figure 3f). No differences were observed for the other three subsets (Figure 3f). 

### 2.5. CD83 Expression in SLN LCs of Patients with Melanoma after In Vitro Stimulation with Inflammatory Cytokines 

Subsequently, we investigated whether LCs from melanoma SLNs were still able to reach a mature immunophenotype with CD83 up-regulation by fluorescent immunocytochemistry and flow cytometry (*n* = 5 negative SLNs from *n* = 5 patients) (Figure 4). Fresh SLN, 24 h-unstimulated and -stimulated SLN cells were labelled with anti-Langerin and CD83 Abs with cell membrane permeabilization, in order to detect all SLN LCs (see Material and Methods section for details). Among fresh SLN Langerin^+^ LCs, characterized by both surface and cytoplasmic labelling, only few LCs expressed CD83 molecule, as revealed by fluorescent immunocytochemistry (Figure 4a): co-expression of both markers revealed the strong prevalence of Langerin on CD83 staining (Figure 4a, merge panel). No substantial differences were observed in SLN LCs cultured for 24 h without inflammatory cytokines as compared to *t* = 0 (data not shown). After 24 h-stimulation with inflammatory cytokines, we observed a strong CD83 up-regulation in SLN LCs (Figure 4b): many SLN Langerin^+^ LCs expressed CD83 with similar intensity (Figure 4b, merge panel). Furthermore, some LCs displayed only cytoplasmic Langerin expression, as further indication of maturation (Figure 4b, merge panel). By flow cytometry, we found that CD83 expression significantly increased in SLN LCs after 24 h-stimulation with inflammatory cytokines (Figure 4f), as compared to both fresh SLN LCs (Figure 4d) and 24 h-culture without inflammatory cytokines (data not shown) (59.15%, range 47.11–77.25%; 19.29%, range 13.26–23.17%; and 19.94%, range 15.47–24.42%, respectively; *p* < 0.05), suggesting that SLN LCs are not irreversibly impaired by melanoma cells and immunosuppressive soluble factors. Inflammatory cytokines stimulation induced a strong IDO1 upregulation by the majority of all LCs in accordance with previous report [31].

## 3. Discussion

In this study, we investigated the expression of the tolerogenic enzyme IDO1 by LCs in SLN of patients with melanoma in order to understand their role in tolerance towards melanoma. We also correlated IDO1 expression with the DC maturation marker CD83, and evaluated SLN LCs functional properties. LCs were located in the T cells-areas of SLN and exhibited the typical dendritic appearance, as observed in healthy LN [9] and in contrast to previous report describing morphological alterations of DC in melanoma SLN [6].

A portion of SLN LCs expressed IDO1, in both negative (3.63%,) and positive (4.18%), SLNs with no significance differences. Approximately 50% of SLN LCs expressed CD83, according to previous reports [11,12]. Based on the expression of IDO1 and/or CD83 by LC, four subsets were identified, in both negative and positive SLN. IDO1^−^CD83^+^ LCs (really mature), IDO1^−^CD83^−^ LCs (real immature), a phenotype corresponding to that of immature skin LCs (Di Gennaro et al., 2014). A further LCs subset co-expressed IDO1 and CD83 (tolerogenic mature), already described in skin LCs and probably with a regulatory role in promoting T cell tolerance [28,29]. Finally, a fourth LCs subset was IDO1^+^CD83^−^ (tolerogenic immature). Notably, this subset was significantly higher in positive SLNs as compared to negative ones and in SLN from melanoma with intermediate/high MR. These findings suggest a role for IDO1^+^CD83^−^ LCs in favoring metastases arrival/homing in LNs and a relation with the proliferation activity of melanoma. The CD83 down-regulation and IDO1 expression might be induced by melanoma and represent a novel immuno-escape mechanism. However, CD83 down regulation on LCs is not definitive. After in vitro 24 h-stimulation with inflammatory cytokines, LCs are able to up-regulate CD83 expression. This finding is particularly important because CD83 down-regulation appears to be crucial for correct T cell activation and proliferation [12] and opens additional perspectives in the treatment of melanoma. Beside inflammatory cytokines, other immune adjuvants might be used to activate LCs and other DC subsets, at tumor site or in the SLN, such as Toll-Like Receptors (TLRs) agonists [30]. In line with this observation, peri-tumoral injection of GM-CSF prior SLN biopsy in melanoma patients reduces SLN immunosuppression [8].

In the last years, numerous reports of immuno-phenotypic and functional defects of SLN DCs in cancer patients have pointed out the concept of “tolerogenic DC”, particularly in melanoma [6,7,11,12]. LCs in SLN of patients with melanoma shows an immature immunophenotype, with low CD83 expression [11]. Since CD83 molecule is important in the stimulation of T cell proliferation [32], it is not surprising that LCs resulted to be poor activator of T cells proliferation [12]. Thus, LCs may be ineffective in stimulating melanoma-specific T cells promoting tolerance instead of anti-melanoma immunity. Furthermore, in patients with melanoma, pDCs showed functional defects and accumulated in metastatic SLN [10]. It has been reported that pDCs and dDCs express the tolerogenic enzyme IDO1 in SLN of patients with melanoma (5,19,20), suggesting a crucial role of this enzyme in melanoma tolerance [13]. Here we showed that also LCs expressed IDO1, although IDO1^+^ LCs were not as numerous as SLN IDO1^+^ pDCs and no difference in frequency between negative and positive SLNs was detected. Importantly, a negative prognostic role of IDO1^+^ cells in SLN of patients with melanoma has been reported [13] according to previous hypotheses that IDO1 expression represents a melanoma immune escape mechanism [13,14]. Besides IDO1 induction in peri-tumoral DCs, melanoma cells themselves can express IDO1 [25,26,27]. which is induced in vitro by interferon-gamma [26]. Importantly, the level of IDO1 expression correlates with a poor prognosis. High levels of IDO1 expression have been detected in primary and metastatic LNs of patients with poor survival, while it was not detected or detected at low levels in patients with long survival [26]. Furthermore, IDO1 expression in primary cutaneous melanoma correlates with Breslow thickness, and IDO1 expression in antigen-presenting cells correlates negatively with progression-free survival of patients with melanoma [27]. It is likely that pro-inflammatory cytokines generated by the innate immune system in response to melanoma may indeed favor IDO1 expression in both melanoma cells and peri-tumoral DCs [1,25]. In our study, the observed small-size IDO1^+^ cells might correspond to dDCs [22] and pDCs [10], while the large-size, observed in positive SLNs, may represent melanoma cells. 

Notably, IDO1 inhibitors have already entered clinical trials, combined first with chemotherapeutic drugs and then with immune check-point inhibitors, such as anti-CTLA-4 and/or PD-1, to turn melanoma tolerance into anti-melanoma immunity [24].

Recently, a novel role for IDO1 as an inflammatory modifier has been described, being the key factor of several pro-inflammatory pathways in cancer responsible for disease progression [25]. Furthermore, IDO1 is involved in tumor neo-vascularisation, which is instead inhibited by IFN-gamma [25]. Thus, IFN-gamma shows a conflicting role in tumor immunity, being the main IDO1 inducer but also involved in preventing new blood vessels formation during tumor growth [33]. Furthermore, tumor rejection mediated by CD8^+^ effector T cells is preceded by the inhibition of tumor-induced angiogenesis by IFN-gamma [33]. Therefore, IDO1 inhibitors might act also as immune adjuvants able to reprogram the correct inflammatory response and block tumor neo-vascularisation [24,25]. Human skin LCs express functional AhR, the well-known target of KYN, responsible for the main IDO1 suppressive actions [34]. Moreover, AhR stimulation by KYN leads to enhance IDO1 expression in LCs, without inducing CD83 expression [34]. Therefore, a possible scenario in melanoma, DC, and IDO1 interaction may be as follows: inflammatory cytokines produced by innate immunity in response to melanoma may modulate CD83 and IDO1 expression on LCs and generate CD83^+^ LCs, either IDO1^+/−^ which migrated to SLN (real mature SLN LCs and regulatory/inhibitory SLN LCs). IDO1, expressed by both LCs and melanoma cells, lead to the production of KYN, which directly stimulate LCs to express more IDO1 trough AhR without inducing CD83 expression. Thus, AhR stimulation in addition to immunosuppressive tumor cytokines might be responsible for CD83 downregulation and generation of CD83^−^ IDO1^+^LCs (immature tolerogenic LCs) and/or CD83^−^ IDO1^−^LCs (real immature LCs).

Importantly, SLN IDO1^+^ DC subsets (LCs, pDCs and dDCs) might be responsible for TRP depletion and KYN production, blocking effector T cells and enhancing T regulatory cells, therefore maintaining a tolerance milieu favoring SLN melanoma metastases [22].

The presence of four SLN LCs subsets reported here might contribute to improving the understanding of LCs biology in melanoma patients. Further studies might elucidate whether CD83 and IDO1 expression identify actually different SLN LCs subsets or different evolution profiles of the same subset, with different roles in immune responses. These results suggest a tolerogenic role of IDO1^+^CD83^−^ SLN LCs in patients with melanoma, being significantly increased in SLN with metastases and in SLN in patients with intermediate/high MR of the primary lesions.

LCs, the first DCs interacting with cutaneous melanoma, are crucial regulators of immune response, representing a promising target for cancer immunotherapy. These findings offer new inputs to strategies targeting LCs directly at the site of primary melanoma or in SLN.

## 4. Materials and Methods

### 4.1. Human Samples

SLN from melanoma patients undergoing SLN biopsy were collected at the Plastic and Reconstructive Surgery Unit, Regional Melanoma Referral Centre and Melanoma & Skin Cancer Unit, Florence, Italy, after obtaining informed written consent. The study was conducted according to the 1964 Helsinki declaration, and its later amendments or comparable ethical standards, and Local Institutional Ethics Committee approval. After pre-operative lymphoscintigraphy and Patent-Bleu injections, SLN was detected and excised using computer-assisted gamma probe with adjustable collimation as previously reported [35]. For research purposes, SLN cell harvesting and processing were performed as previously described [36] with minor modifications [10]. Briefly, SLN was bisected crosswise by the pathologist and only one cutting surface was scraped 10 times with a scalpel blade. SLN cells were collected in a 15 mL tube by blade rinsing with PBS (EuroClone, Whetherby, UK) and used without further enzymatic treatments, to avoid DC activation and maturation prior to analysis [32], either for fluorescent immunocytochemistry, deposed on slide by centrifugation (cytospins), in vitro stimulation with inflammatory cytokines, or flow cytometry analyses. Due to this peculiar collecting method, SLN cells were few, thus we performed experiments on subgroups and not on all specimens. SLNs were processed for routine histology with hematoxylin/eosin and immunohistochemistry. For intra-surgery histological analyses, only for *n* = 5 patients, half SLN was embedded in OCT-like medium (Killik; Bio-Optika, Milan, Italy) and snap-frozen. Cryostat sections (10 μm) and cytospsins were fixed for 10 min in cold acetone (Sigma, Milan, Italy) at 4 °C, air dried at room temperature and stored at −20 °C until labelling.

### 4.2. Antibodies

The following mouse anti-human IgG1 Abs were used: anti-Langerin (CD207; DCGM4), unconjugated and PE-conjugated, from Immunotech (Marseille, France); CD3 V450-conjugated (UCHT1) from BD-Pharmingen (San Diego, CA, USA); CD19 FITC-conjugated Abs (HD37), from Chemicon (Hampshire, UK); CD83 (HB15e), FITC and APC-conjugated from BD-Pharmingen and BioLegend, respectively (San Diego, CA, USA); IDO1 (700838), Alexa Fluor (AF) 488-conjugated from R&D Systems (Minneapolis, MN, USA). Polyclonal (sheep) anti-human IDO1 was from HBT (Uden, The Netherlands). The following secondary Abs were used: goat anti-mouse AF594-conjugated, from Molecular Probes (Life Tecnologies, Milan, Italy); donkey anti-sheep FITC-conjugated, from AbD Serotec (Oxford, UK). Fluorescein signal was amplified with anti-FITC AF488-conjugated Abs (Molecular Probes). Isotype-matched Abs were used as negative controls.

### 4.3. In Vitro Stimulation with Inflammatory Cytokines

SLN cells were re-suspended in RPMI 1640 medium (EuroClone) supplemented with 1% penicillin/streptomycin (EuroClone) and 2% glutamine (Sigma), without serum, indicate as culture medium. For in vitro stimulation analyses, cells from each SLN were divided into three aliquots: the first was processed immediately, indicated as t = 0; the second and the third were cultured for 24 h in culture medium at 37 °C and 5% CO2 with or without the following inflammatory cytokines: TNF-alpha (10 ng/mL), IL-1beta (10 ng/mL) and IL-6 (1000 U/mL), all from R&D Systems (Minneapolis, MN). Cells were used for fluorescent immunocytochemistry and flow cytometry.

### 4.4. Fluorescent Immunohistochemistry and Immunocytochemistry

Multi-color fluorescent labelling procedures were performed on SLN sections and cell cytospins, at room temperature if not indicated. For LC localization within different SLN areas, three-color fluorescent immunohistochemistry was performed on SLN sections, pre-treated with 20 mg/mL BSA (Sigma-Aldrich, St. Luis, MO, USA) for 1 h, and stained first with anti-Langerin for 2 h andrevealed with AF594-conjugated Abs (red fluorescence); then with anti-CD3 V450- and CD19 FITC-conjugated Abs (blue and green fluorescence, respectively) overnight at 4 °C; next, fluorescein signal was amplified. To study IDO1 and CD83 expression in SLN LCs, two-color fluorescent immunocytochemistry was performed on cytospins, pre-treated with PBS/Triton X-100 0.3% and 20 mg/mL BSA (all from Sigma-Aldrich) for 1 h for cell membrane permeabilization. For Langerin/CD83 labelling, specimens were first stained with anti-Langerin for 2 h, revealed with AF594-conjugated Abs (red fluorescence); then with anti-CD83 FITC-conjugated Abs (green fluorescence), overnight at 4 °C; next fluorescein signal was amplified. For Langerin/IDO1 labelling, specimens were first stained with polyclonal anti-IDO1 overnight at 4 °C, revealed with anti-sheep FITC-conjugated Abs and fluorescein signal was amplified (green fluorescence); then with anti-Langerin for 2 h, revealed with AF594-conjugated Abs (red fluorescence). This labelling order was set up to avoid sheep–goat immune cross-reactions. Nuclei were finally labelled with Hoechst 33,342 (20 μg/mL; Sigma; blue fluorescence). All slides were then mounted with Prolong antiFade (Life Technologies, Thermo Fisher Scientific Inc, Massachusetts, MA, USA.) and observed with Leica DMLB microscope (equipped for epifluorescence; Leica Microsystems GmbH, Wetzlar, Germany). Images were acquired at 0.529 and 0.265 µm per pixel resolution, corresponding to ×200 and ×400 original magnification respectively, using Leica DFC200310 FX microscope digital color camera and LAS software with overlay module (Leica Microsystems GmbH). Adobe Photoshop CS2 software (Adobe Systems Incorporated, San Jose, CA, USA) was used for image processing and figure creating.

### 4.5. Flow Cytometry Analyses

All Abs dilutions and washing steps were done in PBS supplemented with 1% FCS at room temperature if not indicated. For IDO1 and CD83 expression analyses, two- or three-color cell staining was performed on fresh cells and/or after 24 h-culture with or without inflammatory cytokines cells, as previously reported (Gerlini et al., 2012; Di Gennaro et al., 2014). Briefly, for Langerin/CD83 labelling, cells were first labelled with anti-Langerin-PE and CD83-APC for 15 min; then fixed and permeabilized with Cytofix/Cytoperm solution (PharMingen, San Diego, CA, USA) following manufacturer’s instructions, and again labelled with anti-Langerin-PE for 15 min, in order to detect all SLN LCs, because during maturation, surface Langerin is relocated to the cytoplasm (Gerlini et al., 2012). For Langerin/CD83/IDO1 labelling, cells were labelled first with anti-Langerin-PE and CD83-APC as described above; then fixed and permeabilized with Flow Cytometry Fixation Buffer and Permeabilization/Wash Buffer I, all from R&D Systems, following manufacturer’s instructions, and labelled with anti-IDO1 AF488-conjugated for 30 min and Langerin-PE, to reveal cytoplasmic Langerin as reported above (Di Gennaro et al., 2014). Cells were then acquired using FACS Canto and FACSDiva software, version 6.0 (Becton Dickinson Immunocytometry Systems, San Jose, CA, USA). Results were expressed as percentage of total events or CD83^+^ or CD83/IDO1 double positive cells among total Langerin^+^ cells. 

### 4.6. Statistical Analysis

Values reported throughout the text are expressed as mean and values range. Data were analyzed using two-sided student’s *t*-test with *p* < 0.05 adopted as significance level (Origin and Microsoft Excel software).

## Figures and Tables

**Figure 1 ijms-23-03441-f001:**
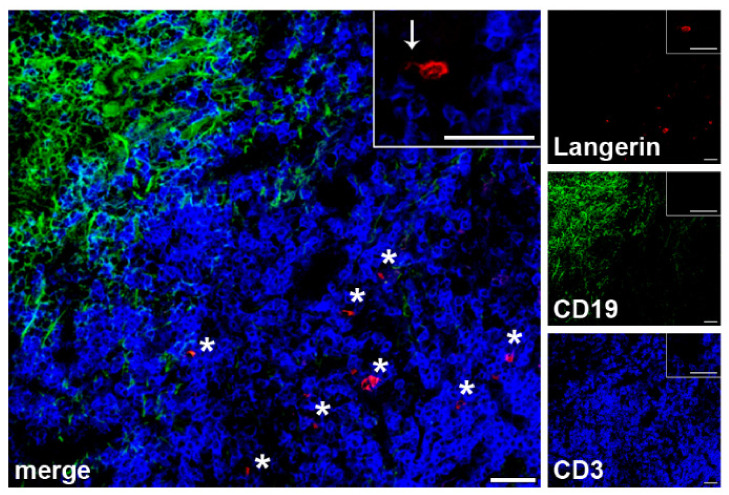
LCs localization within SLN areas of melanoma patients. Three-color immunofluorescence labelling was performed on frozen SLN sections (*n* = 4 negative and *n* = 1 positive SLNs from *n* = 5 patients) with the indicated Abs and fluorescences (anti-Langerin, red; CD19, green; CD3, blue) in order to localize LCs within specific SLN areas. A representative negative SLN is shown. Original magnification ×200 and ×400 in the inset; scale bars = 20 μm and 40 μm, respectively. SLN LCs are found in the T cells rich areas, as indicated by asterisks; a typical SLN LC, with typical thin and long dendrites, is shown in the inset, as indicated by an arrow.

**Figure 2 ijms-23-03441-f002:**
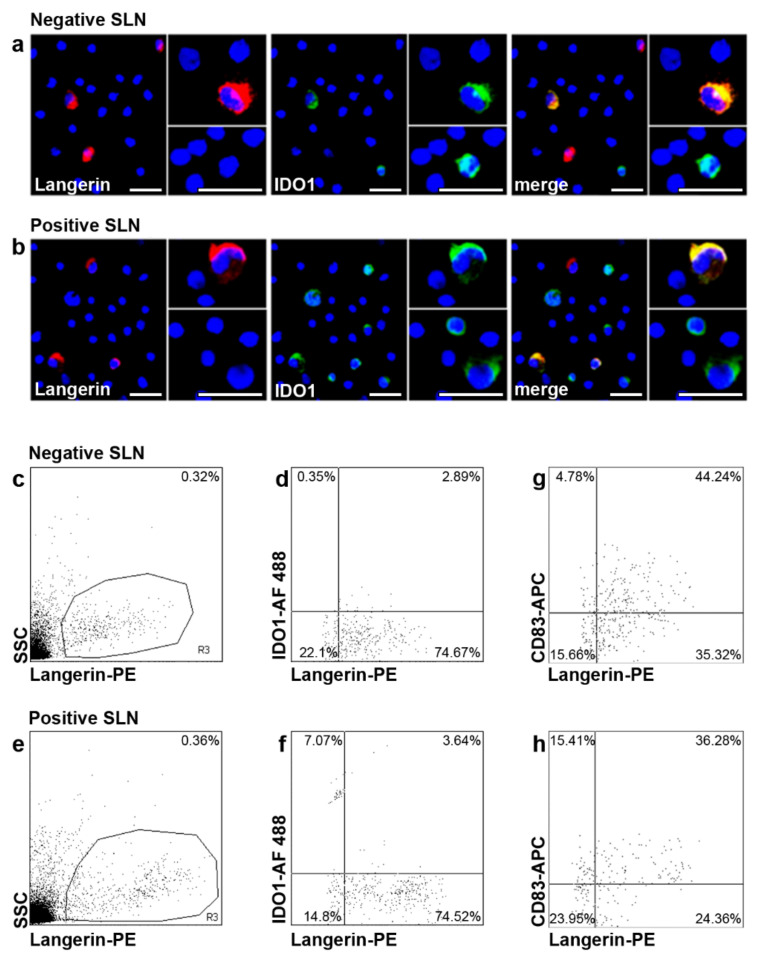
Fluorescent immunocytochemistry and flow cytometry analyses of IDO1 in SLN LCs. (**a**,**b**) Double fluorescent immunocytochemistry analyses were performed on cytospins of negative and positive SLNs (*n* = 10 negative and *n* = 4 positive SLNs from *n* = 12 patients) with the indicated Abs and fluorescences (anti-Langerin, red; anti-IDO1, green); co-expression of both molecules (yellow-orange) is indicated as “merge”. Nuclei were labelled with Hoechst (blue). Original magnification ×200 and ×400 in the inset; scale bars = 20 μm and 40 μm, respectively. Details from representative images from negative and positive SLN are shown. Only a fraction of Langerin^+^ LCs expressed IDO1 (**a**,**b**). Langerin-IDO1^+^ cells were also observed, characterized by different sizes, especially in positive SLNs as compared to negative ones. Original magnification ×200 and ×400 in the inset. (**c**–**h**) Flow cytometry analyses of IDO1 expression in SLN LCs (*n* = 21 negative and *n* = 5 positive SLNs from *n* = 22 patients) were performed with the indicated antibodies and fluorochromes. Results are expressed as percentage of total events or double positive cells among total Langerin^+^ cells. Representative negative and positive SLNs are shown. (**c**,**e**) LCs are present both in negative and positive SLN. (**d**,**f**) Quantitative analyses confirmed that among Langerin^+^ SLN LCs, only few expressed IDO1, with a slight increase in positive SLN as compared to negative ones, although not significant. (**g**,**h**) The percentages of Langerin^+^ CD83^+^ cells are shown in negative (**g**) and in positive (**h**) SLN.

**Figure 3 ijms-23-03441-f003:**
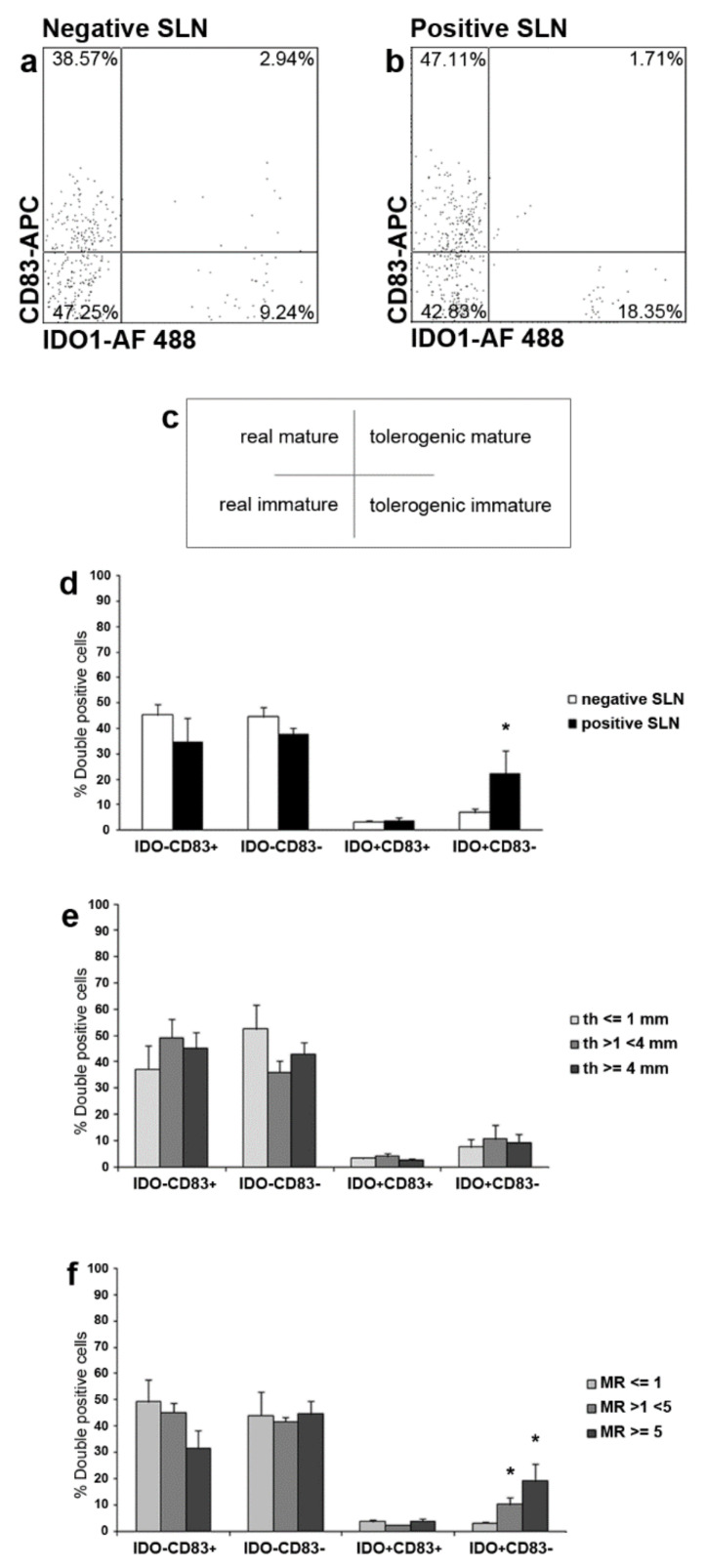
Flow cytometry and statistical analyses of IDO1 and CD83 simultaneous expression in SLN LCs from melanoma patients: identification of four subsets. (**a**–**f**) Three-color flow cytometry analyses were performed to investigate the concurrent expression of IDO1 and CD83 in SLN LCs from negative and positive SLNs (*n* = 21 negative and *n* = 5 positive SLNs from *n* = 22 patients) with the indicated Abs and fluorochromes. (**a**,**b**) Representative negative and positive SLNs are shown. Gated Langerin^+^ SLN LCs were analyzed for both IDO1 and CD83 expressions, thus identifying four SLN LCs subsets as shown in the cross inset (**c**): IDO1^−^CD83^+^, indicated as real mature LCs (upper left); IDO1^−^CD83^−^, indicated as real immature LCs (lower left); IDO1^+^CD83^+^, indicated as tolerogenic mature LCs (upper right); and IDO1^+^CD83^−^, indicated as tolerogenic immature LCs (lower right). Values indicate SLN LCs subsets percentages among total SLN LCs. (**d**–**f**) Quantitative and statistical analyses of SLN LCs subsets are shown; (*) indicates *p* < 0.05; percentage of double positive cells frequencies is displayed on y-axes. The real mature and immature SLN LCs subsets appeared lower in positive SLN as compared to negative one, although differences were not significant. The tolerogenic mature SLN LCs subset was almost invariant between positive and negative SLN. On the contrary, the tolerogenic immature SLN LCs subset was significantly higher in positive SLNs as compared to negative ones. (**e**) Taking into account the Breslow’s thickness (th) of the primary lesions, we did not observe differences in the four SLN LCs subsets from patients with thin (<1 mm), intermediate (>1 and >4 mm) and thick (≥4 mm) primary melanomas. (**f**) In contrast, concerning the mitotic rate (MR) of primary lesions, a significant increase was observed for the tolerogenic immature subset in SLN LCs from patients with higher MR (>1 and <5, and ≥5), as compared to the same subset from patients with MR ≤ 1.

**Figure 4 ijms-23-03441-f004:**
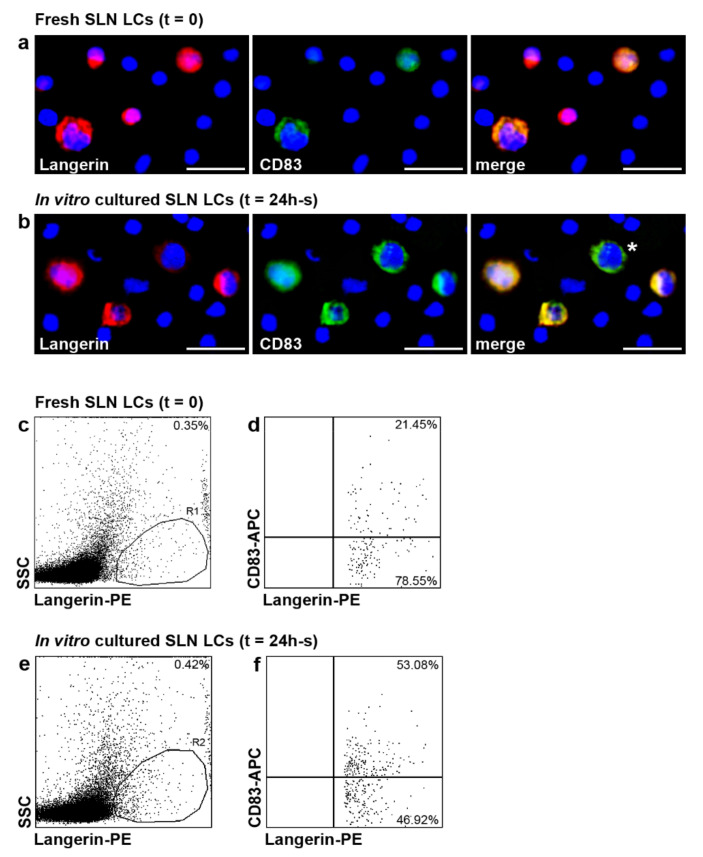
CD83 expression increased in SLN LCs from melanoma patients after in vitro stimulation with inflammatory cytokines. (**a**–**f**) CD83 expression was analyzed in fresh and 24 h-stimulated with inflammatory cytokines SLN LCs (indicated as *t* = 0 and *t* = 24, respectively), by immunofluorescence (**a**,**b**) and flow cytometry (**c**–**f**) analyses. A representative SLN, fresh and after stimulation with inflammatory cytokines, is shown (*n* = 5 negative SLNs from *n* = 5 patients). (**a**,**b**) Double immunofluorescence labelling was performed on SLN cells deposed on cytospins with the indicated Abs (anti-Langerin, red; CD83, green); nuclei were labelled with Hoechst (blue) and co-expression of both molecules (yellow) is indicated as “merge”. Original magnification ×400; scale bar = 40 μm. (**a**) At *t* = 0, only few SLN LCs expressed CD83, while in 24 h-s cultures with inflammatory cytokines (**b**), many SLN LCs expressed CD83 and some of them also reduced Langerin expression (indicated by asterisk). (**c**–**f**) Two-color flow cytometry analyses were performed on permeabilized SLN cells suspension with the indicated Abs and fluorochromes to detect all SLN LCs. Results are expressed as percentage of Langerin^+^ cells (R1 and R2, respectively) among total events (**c**,**e**), and double positive cells among total Langerin^+^ cells (**d**,**f**). CD83 expression significantly increased in SLN LCs (**f**) after in vitro stimulation with inflammatory cytokines, as compared with fresh SLN LCs (**d**).

**Table 1 ijms-23-03441-t001:** Clinical features of patients with melanoma.

Patients		Positive SLN	Negative SLN	Total SLN
Stage III	8	10		
Stage I–II	36		40	
total	44			50
Sex				
Men	20	4	16	
Women	24	4	20	
Age (mean, years)		62 years range 27–80	55 years range 21–93	
Breslow thickness (mean, mm)		4.19 mm range 1.1–9.8	2.18 mm range 0.5–8.1	
Lesion Type *				
SSM	30	4	26	
NM	12	4	8	
ALM	2		2	
Total	44	8	36	
Mitotic rate (mean)		5.2 (2.0–12)	1.3 (0.0–9)	
Mean interval ** (mean, days)		39 days range 0–61	41 days range 0–71	

(*) SSM, superficial spreading melanoma; NM, nodular melanoma; ALM, acral lentiginous melanoma. (**) Mean interval between excision of primary melanoma and surgery for enlargement and sentinel node biopsy. VIII ed. AJCC (American Joint Committee on Cancer).

## Data Availability

Data supporting reported results can be available at our Institute.

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
