# Peer review of "Tolerogenic IDO1+CD83− Langerhans Cells in Sentinel Lymph Nodes of Patients with Melanoma"

_ijms, 2022, doi:10.3390/ijms23073441_

Round 1
Reviewer 1 Report
The authors are presenting the results of a comprehensive analysis of melanoma patients using complementary assays: Fluorescent Immunohistochemistry, Immunocytochemistry and Flow cytometry analyses. All the methods are described in details. The results are sustained by suggestive images.
- Line 51: please define enigmatic subset of cells
In patients with melanoma, IDO1 expression were first observed in an enigmatic subset of cells with plasmacytoid shape in SLN [18,19].
- The research described in this manuscript is too much detailed for me, to be able to criticise. Still, I do not understand why the Dr which performed all the experiments presented in this paper is only mentioned in the Acknowledgements and is not one of the authors. For me, which I am lab person, it will be more reliable that the results to be explained by the person who made those experiments, and not from surgeons and clinicians.
Let us hope that these results will provide new input for immunotherapeutic approaches targeting in vivo LC of patients with melanoma.
Reviewer 2 Report
This manuscript investigates IDO1 expression on immature and mature Langerhans cells in sentinel lymph nodes of melanoma patients.
There are several major and minor concerns that I would like to see addressed, which are detailed below.
Major concerns:
1) Langerhans cells are referred to as a Dendritic cell subset throughout the manuscript, while current literature classifies these cells as macrophages instead. Please justify this alternative classification or revise the manuscript.
2) Although a total of 49 SLNs from 43 melanoma patients were collected, the results refer to using 14 SLNs from 12 patients, 26 SLNs from 22 patients, or 5 SLNs from 5 patients. It is unclear how these subgroups were selected and why these studies were not performed on all specimens.
3) Scale bars are missing on all immunofluorescent images (Figures 1, 2a-b, 4a-b) and cells shown in magnified inset images cannot be seen in lower magnification paired images. Also, no quantification of detected double or single positive cell counts across all specimens was performed, despite several statements pertaining to cell subset abundance.
4) The representative FACS plots of langerin+IDO1+ cells (Figures 2d and f) and CD83+/-IDO1+ cells (Figures 3a and b) show very small numbers of cells, which are not very convincing. The gating strategy used here should be included as supplementary figures, including Fluorescence Minus One (FMO) controls.
5) The local ethics approval name and reference number should be included in the Institutional Review Board Statement.
Minor concerns:
6) All Figures are listed at the end of the results section, instead of when mentioned in the text, please revise.
7) IDO1 is at times referred to as IDO throughout the manuscript, please revise.
8) Section 2.1, page 2: Please include a Table summarizing the clinical data and include information regarding Breslow thickness and mitotic rate.
9) Section 2.2, page 2, and Figure 1: Thin long dendrites are not clearly seen in Figure 1, a nuclear stain should be included to confirm positive LC staining, and the figure legend is incorrectly formatted as the main text.
10) Section 2.3, page 3, lines 97-99: Please include these data as a main or supplementary figure.
11) Section 2.4, page 3, lines 124-129: Figure 3f is incorrectly mentioned as 3e, please revise.
12) Figures 3 d, e and f are missing axes labels, please revise.
13) Discussion: CD83 is not only a DC maturation marker, please revise. It would have been interesting to investigate the ability of the CD83+ LCs to induce T cell proliferation, particularly after cytokine-induced enhanced expression of CD83.
14) Section 4.6, page 11: Please include which software was used for statistical analysis.
15) References, page 13: please remove reference 38, which is blank and not included in the text.
Round 2
Reviewer 2 Report
I thank the authors for addressing my concerns and editing the manuscript accordingly.
I have no further comments.